# Understanding Why All Types of Motivation Are Necessary in Advanced Anaesthesiology Training Levels and How They Influence Job Satisfaction: Translation of the Self-Determination Theory to Healthcare

**DOI:** 10.3390/healthcare9030262

**Published:** 2021-03-01

**Authors:** Parisa Moll-Khosrawi, Stefan Zimmermann, Christian Zoellner, Leonie Schulte-Uentrop

**Affiliations:** 1Department of Anaesthesiology, University Medical Center Hamburg-Eppendorf, Martinistr. 52, 20246 Hamburg, Germany; c.zoellner@uke.de (C.Z.); lschulte@uke.de (L.S.-U.); 2Institute of Biochemistry and Molecular Cell Biology, University Medical Center Hamburg-Eppendorf, Martinistr. 52, 20246 Hamburg, Germany; stezim84@googlemail.com

**Keywords:** self-determination, motivation, job-satisfaction

## Abstract

Studies applying the self-determination theory have shown that intrinsic motivation and autonomous regulation lead to job satisfaction and to better job performance. What has not been worked out clearly yet are the effects of extrinsic motivation and controlled regulation on affect, job performance and job satisfaction. However, it has been described that controlled regulation is often necessary for mundane tasks. In anaesthesiology, routine daily tasks can be perceived as mundane by those who have achieved a certain level of training (e.g., consultants). Therefore, it was hypothesised that consultants have high expressions of all motivational qualities. Furthermore, it was hypothesised that job satisfaction of anaesthesiologists is correlated with autonomous motivation. The hypotheses were tested in a cross-sectional study design within a group of anaesthesiologists. The study participants reported the same pattern throughout the motivational continuum. Consultants reported the highest levels of all motivational qualities, including controlled regulation, as well as the highest levels of job satisfaction. Junior residents reported high levels of amotivation and extrinsic regulation. The lowest levels of identified regulation and job satisfaction were reported by the group of attendings. Job satisfaction was positively correlated with intrinsic motivation and negatively correlated with amotivation. Therefore, our findings from the field of anaesthesiology show that the expressions of high levels of controlled regulation might be necessary for specialists to engage in mundane daily tasks. Intrinsic motivation and autonomous regulation are necessary for job satisfaction and the presence of controlled regulation and extrinsic behavioural regulation have no declining effects. Furthermore, the decrease of amotivation will lead to enhanced job satisfaction and the resulting consequences will be extensive. Junior residents need to be supported with the aim to enhance their feeling of autonomy and competence in order to decrease amotivation and to foster autonomous regulation and hence to increase job satisfaction and well-being. Further special focus should be on attendings to counteract their lacking identification with the job. Hereby, the provision of feedback and professional perspectives might foster the process of re-identification.

## 1. Background

Many medical departments are facing challenges in providing job satisfaction for residents, combined with a threatening lack of doctors [1,2]. In several fields as well as in healthcare, investigations showed a positive correlation between motivation and job satisfaction, as well as performance of employees [3] and, therefore, motivation plays a crucial role to provide efficient and good quality patient care [4].

In the early 20th century, it was assumed that employees’ motivation was solely monetary based [5]. This assumption was withdrawn after the Hawthorne studies showed that employees changed their working behaviour and productivity when they were observed. Subsequent research revealed and confirmed that employees are motivated by intrinsic and extrinsic motivators [2].

Whereas other motivational theories of human behaviour mainly focus on different quantities of motivation, the self-determination theory (SDT) [6], one of the leading motivational theories describes that different types of motivators result in different qualities of motivation with varying outcomes, detached from the quantity of motivation [7]. In order to study motivation in complex work settings like healthcare or anaesthesiology, where the point of concern is not only if the work is done and how fast it is done, but much more how it is done and how the employees connect to their work, this differentiated view from the SDT on motivation and its qualities is necessary. Furthermore, the SDT focusses substantially on autonomous motivation and is the only theory that has deeply explored autonomy, using empirical methods [8]. Therefore, the SDT is the appropriate approach to explore motivation and motivational patterns in healthcare employees.

SDT is based on the postulation that humans have an innate will to grow and this tendency to grow can be supported or hampered by intrinsic or extrinsic factors as well as situations. The psychological growth of human beings is determined by the satisfaction of three basic psychological needs: Autonomy, competency and relatedness.

Other than in the motivation theory of Porter and Lawler, who have a dichotomous description of motivation in intrinsic and extrinsic, SDT specifies motivation on a scale and describes different forms that can guide individual behaviour [6,9]. Every motivation underlies a type of regulation, regulatory process and locus of causality.

When an activity is carried out due to inherent satisfaction, usually intrinsic motivation is present-SDT associates this with cognitive and social development. Therefore, intrinsic motivated individuals are autonomously regulated. When an activity is conducted due to external sources, such as punishment or reward, extrinsic motivation is foregrounded. Four types of extrinsic motivation are described in SDT, which vary in terms of their relative autonomy [6]: *External regulated* behaviour is least autonomous, it is only based on demands or punishment or possible rewards [10]. *Introjected regulation* is more autonomous than external regulation, but still the activity or rule is seen as conditioned by others [10,11]. When it comes to avoid guilt and attain self-esteem and citing ego, the predominant regulation is of introjected nature. Introjected regulation is predominant to avoid guilt, attain self-esteem and citing ego [6,11,12,13]. Moving on the continuous motivational scale, *identified regulation* is more autonomous than introjected regulation: here the action is accepted as important and involves consciously valuing a goal [13,14]. When regulations are connected to oneself, the most autonomous form of extrinsic regulation, *integrated regulation* occurs [10,11,13]. Integrated regulation shares qualities with intrinsic motivation, although classified as extrinsic. Intrinsic, integrated and identified regulation are summarized as “autonomous self-regulation”, whereas extrinsic and introjected regulation are summarised as “controlled self-regulation”.

Having no motivation at all is referred to as amotivation, which can occur if a person experiences lack of competence or does not see the reason for a task [6,15].

It has been demonstrated that intrinsic motivation is associated with the satisfaction of the three basic needs [16,17]. Several studies revealed that rewards may even impair intrinsic motivation and enhance extrinsic motivation. Some investigations doubt this phenomenon and state that rewards do not influence external regulation [18,19,20].

Studies in different fields like physical exercise, education, health care and the work place have shown that autonomous self-regulation lead to academic achievement [21], positive emotions [22], perceived competence and self-worth [23], creativity [24], retention and less drop out of activities or school attendance [15,25], better performance on complex tasks [26] and correlated with greater job satisfaction and well-being [14,27,28,29,30,31]. In addition, employees who were autonomously regulated were less at risk for burn-out [32]. In the work place, controlled self-regulation was associated with more turnover and burn-out [3,33].

The basic psychological needs cannot be satisfied by all means in the work setting and intrinsic motivation is less likely to occur at the work place than in hobbies [16]. Work motivation is largely influenced by the social context in which employees operate and their motivation is not only affected by resources but also by job demands [34]. Autonomy supportive environments at work lead to better well-being and enhanced autonomous self-regulation [17,27,35] and decision making autonomy minimized negative effects of job demands [32]. Based on its positive effects, enhancement of autonomous motivation should be a key concept for healthcare professionals [14], but what if the natural job characteristics foster different behavioural regulations? As an example, it is conceivable that certain high positions in health care come along with higher needs to attain self-esteem and citing ego, then, the predominant regulation would be introjected (controlled). On the other hand, especially in high risk workplaces like anaesthesiology or aviation, the routine everyday activities can underchallenge experts and this could also influence their motivation. Knowledge about how motivation of employees is affected by job characteristics and training levels is important, because motivation has effects on well-being and job satisfaction [36] and should be considered as a key concept to maintain well-being [6].

It is appropriate to place special emphasis on health care employees’ well-being, because physicians worldwide and across all specialties report burnout and stress [37]. Some authors even outline anaesthesiologists to be at greater risk to develop burnout and decreased mental well-being [38,39]. Internationally, the burnout risk for anaesthesiologists is estimated at around twenty-five to fifty percent [40,41,42,43,44]. These numbers are congruent with those reported for European- [45,46,47,48,49] and German anaesthesiologists [50]. The consequences of burnout and low job satisfaction are extensive and not only limited to the employee. Burnout leads to a reduction in patient safety [51,52,53,54] and conversely job satisfaction leads to increased productivity and decreased treatment costs and content patients [55,56,57]. The relation between job satisfaction and burnout has been confirmed [58] and it has been demonstrated that job satisfaction protects against burnout [59,60,61,62,63]. Therefore, the promotion of job satisfaction would result in various positive outcomes on many organisational levels.

So far, some qualitative investigations outlined which general factors and working conditions have enhancing effects on job satisfaction [58,61,62,63]. All these factors comply with the satisfaction of the basic psychological needs: competence, autonomy and relatedness, as postulated by the self-determination theory [6]. Therefore, motivation, job-satisfaction and burnout are integrally related, and the aforementioned investigations need to be complemented with regard of motivational expressions of healthcare employees. Filling this gap in knowledge will provide the basis for the identification of employees who are at special risk and for the correlation of motivational qualities with job satisfaction. Then, motivation can be considered as an indirect surrogate parameter for burnout and job satisfaction and specific preventive actions and interventions can derive from the expanded evidence.

In anaesthesiology, evidence about the course of motivation during residency and in higher positions (attending, consultant) is scarce.

Therefore, the aim of this study was to analyse the distribution of motivation within a sample of different year anaesthesiology residents, attendings and consultants, all facing different working routine and responsibilities and to analyse the correlation between job satisfaction and the different levels of motivation.

It was hypothesised that experts (consultants) of anaesthesiology have high levels of all motivational qualities. Further hypothesis was, that experts (consultants) need high levels of controlled regulation to engage in daily tasks, therefore, in higher training levels, controlled regulation has no declining effects on job satisfaction. Furthermore, it was hypothesised that job satisfaction of anaesthesiologists is correlated with autonomous motivation.

## 2. Methods

### 2.1. Study Design and Setting

This cohort study was performed at the Department of Anaesthesiology in the University Medical Center of Hamburg-Eppendorf, Germany. An email with information about the study was sent in May 2018 to all anaesthesiologists of the department, including a questionnaire which they were asked to fill out and to return anonymously within a time span of two weeks. The questionnaire included a question to assess job satisfaction and a German translated version [64] (see Appendix A) of the Situation Motivation Scale (SIMS) [65], adapted by Gillet and colleagues [20], which measures participants’ situational motivation towards engaging in an activity, at a specific point of time.

### 2.2. Participants

All anaesthesiologists of the department (*N* = 186) were eligible for the study as no specific eligible criteria were necessary and a broad cross-sectional design was chosen.

A detailed description of the departments’ organisational and personnel structure and job descriptions of the investigated subgroups are provided in the Appendix A.

A total of 71 anaesthesiologists (38%) of our department took part in the study and finished the SIMS questionnaire. The reasons for non-participation were not assessed, as participation was voluntary. We assumed that main reasons for non-participation were absence during the study period (e.g., ICU rotation, vacation, scheduling in other OR than in the central OR of the department with distance to the collection box). Table 1 shows the training level of the study participants.

### 2.3. Outcomes

#### Situational Motivation Scale

To avoid cognitive bias, participants were asked not to fill out the questionnaire during or directly after night-shift [36,66].

The participants’ situational motivation towards performing anaesthesiology was measured using a translated version (German) of the Situation Motivation Scale (SIMS) [65], adapted by Gillet et al. [20]. The SIMS has four subscales, measuring the type of motivation to run an activity at a specific point of time [65] and studying the important question why an individual shows a certain behaviour [67]. Consequently, it is possible to compare the motivational measurement with its conceptual definition that refers to the recognised reason of task engagement [15,68,69]. The adapted version of the SIMS has five subscales, with four items per subscale, measuring intrinsic motivation, extrinsic-, identified-, introjected regulation and amotivation. Each item has a 7-point Likert scale (1 = “Does not correspond at all” and 7 = “Corresponds exactly”) and participants were asked to specify the extent to which each item represented a reason why they were performing anaesthesiology.

The motivational qualities (subscale) are computed based on the corresponding items (four items per subscale). [65] As an example, introjected regulation is calculated by adding and averaging the items 5, 10, 15 and 20. (A detailed explanation is provided in the Appendix A).

A computed autonomous motivation index was calculated by adding and averaging the intrinsic motivation and identified regulation, and a controlled motivation index was computed by adding and averaging extrinsic- and introjected motivation [20,65]. The validity and reliability of the SIMS, as well as of the adapted version, have been confirmed in several studies [20,65,70]. We also tested the internal consistency of the adapted German version, calculating Cronbach’s alpha for each subscale. Our results confirm the reliability for each subscale of the German SIMS (intrinsic: α = 0.72; identified: α = 0.68; introjected α = 0.71; extrinsic: α = 0.79; amotivation α = 0.75).

Job satisfaction was assessed by a 7-point Likert scale (1 = “Not satisfied”and 7 = “Completely satisfied”).

There are no specific cut-off values for the subscales of the SIMS, describing if a type of motivation is too low. However, the scores can be interpreted regarding inter-individual differences. Therefore, we did not divide the results in high or low categories and just compared the mean values of motivational subscales between different years of anaesthesiology residency, attendings and consultants.

The same principle was used to analyse the job satisfaction results-the higher the score, the higher the job satisfaction was categorized.

### 2.4. Statistical Analysis

Statistical analysis was performed using IBM SPSS (IBM, Armonk, NY, USA) Statistics Version 23.0. No data was missing in the returned questionnaires. The internal consistency of the adapted German version was analysed, calculating Cronbach’s alphas for each subscale. Mean differences in situational motivation and job satisfaction were compared by year of anaesthesiology residency in an analysis of variance (ANOVA). The significant results of the ANOVA were further analysed (post-hoc) with Bonferroni adjusted correction for multiple testing.

## 3. Results

### 3.1. Situational Motivation

Figure 1 depicts the expression of all motivational qualities for each investigated group of anaesthesiologists. Overall, each group followed the same pattern throughout the motivational continuum. The levels of reported autonomous regulation (*M* = 5.60, *SD* = 0.88) were high- and the levels of controlled regulation (*M* = 2.70, *SD* = 0.92) low within the whole group (Figure 1, Table 2).

The 1st and 2nd year residents reported the lowest levels of intrinsic motivation (*M* = 5.69, *SD* = 1.23), the highest levels of amotivation (*M* = 2.11, *SD* = 1.32), as well as external regulation (*M* = 2.47, *SD* = 1.56).

Our results confirmed our hypotheses that the high training levels of anaesthesiology are accompanied with high levels of all motivational qualities, as well as controlled regulation. The consultants reported the highest expression for all motivational qualities. Their levels of introjected regulation were significantly higher than the introjected levels of the 3rd (1.34, 95%-CI [0.01, 2.77]) and 4th/5th year residents (1.92, 95%-CI [0.57, 3.26]).

Furthermore, the consultants reported significantly higher levels of controlled regulation than the 4th and 5th year residents (1.03, 95%-CI [0.02, 2.04]) (Figure 1, Table 2).

### 3.2. Correlation of Situational Motivation and Job Satisfaction

The highest score of job satisfaction was found within the consultants (*M* = 5.78, *SD* = 0.44), followed by the 1st and 2nd year residents (*M* = 5.64, *SD* = 1.00). The lowest score of job satisfaction was reported by the attendings (*M* = 4.83, *SD* = 1.64).

Table 3 shows the zero-order correlations of all motivational qualities with each other and with job satisfaction.

Our results confirmed our hypothesis that the job satisfaction of anaesthesiologists is correlated with autonomous regulation (*r* = 0.299) and therefore (autonomous regulation is the sum of intrinsic motivation and identified regulation) also positively correlated with intrinsic motivation (*r* = 0.360).

Furthermore, job satisfaction was negatively correlated to amotivation (*r =* −0.27).

Our hypothesis was confirmed, that controlled regulation has no declining effect on job satisfaction.

Other than often described, our results showed a positive correlation of introjected- with autonomous regulation (*r* = 0.26).

## 4. Discussion

In our cross-sectional study, we found that consultants reported the highest expression of all motivational qualities and the levels of introjected- and controlled regulation were significantly higher than reported by the other study participants. The high expressions of controlled- and introjected regulation had no declining effects on job satisfaction—quite the contrary, consultants reported the highest job satisfaction.

Within the whole sample, job satisfaction was positively correlated with intrinsic motivation and autonomous regulation- and negatively correlated with amotivation.

As shown in Figure 1, consultants reported the lowest levels for amotivation but the levels for intrinsic-, identified- extrinsic- introjected-, autonomous and controlled regulation were higher compared to the other study participants. Considering these results, consultants perceive the locus of causality to engage in anaesthesiology related activities, simultaneously combined form the inside and outside (high expression of autonomous- as well as controlled regulation). The explanation for this, at first glance contradictory phenomenon, can be derived from the job characteristics of anaesthesiologists, as well as from the SDT: The high intrinsic and autonomous levels of motivation/regulation are due to consultants’ personal endorsement to engage in anaesthesiology related tasks. Furthermore, consultants have the ability to create their working day autonomously and can therefore influence plenty of working processes (autonomy). Due to their level of training, they are experts of anaesthesiology which is coupled with a broad and final decision- making scope of all medical issues. Hereby, the feelings of autonomy and competence are augmented [71], which in turn results in autonomous regulation and the motivation for the actions are of intrinsic nature [72].

The high levels of controlled- and introjected regulation might be due to an amount of competition which is present amongst the consultants and therefore, the reason for some job activities are to cite ego and to attain pride- as expressions of introjected regulation are high when engagement in a task takes place to avoid guilt, to attain pride and self-esteem [10,11,12,73]. Furthermore, although consultants have the highest expertise and decision-making authority, they are also engaged in daily processes which might be perceived as mundane. It is reported, that for conducting mundane and daily tasks, the existence of controlled regulation leads to better performance [27,74]. Therefore, the high levels of controlled regulation might even be necessary for experts (consultants) to withstand non-challenging and daily tasks.

The positive effects of autonomous regulation have been reported [27,28,29,30,75], whereas the effects of controlled regulation were uncertain for a long period of time [18,76,77].

Gillet and colleagues reported that the manifestation of controlled regulation lead to negative affect, but did not have any influence on performance and job satisfaction [20]. A further study showed that autonomous and introjected regulation can coexist and still be associated with positive job characteristics and better psychosocial well-being, independently from extrinsic regulation [78]. Findings from sport science research highlighted even that athletes’ performance was better if motivation was reported upon the whole continuum [76].

These findings were also confirmed in investigations on job perfectionism and workaholism, which revealed that employees with manifestations of motivation throughout the motivational continuum had higher levels of these features [79,80].

Our results confirm the aforementioned findings that introjected regulation has no declining effect on job satisfaction. The coexistence of controlled and introjected regulation alongside all other motivational qualities might even be the best state in which experts perform.

However, it should not be overlooked that attendings too, have reached a certain expertise due to their finished specialist training. Nevertheless, they are subservient to the consultants in many aspects, including the final decision. Basically, they are waiting for a promotion to take on the role of a consultant. In concordance with our results, this state might cause a lack of identification with the job [81] and a state of discontent, resulting in decreased identified regulation and job satisfaction. Regular autonomy supportive feedback and the demonstration of perspectives might be a first step to circumvent the non-identification and to foster autonomous motivational qualities.

Although one expects Junior residents (1st and 2nd year) to be intrinsically or autonomously motivated, in our study, they reported the highest levels of amotivation and extrinsic motivation. The expectation that junior residents are autonomously motivated is based on the circumstance, that after many years of medical school, they finally can narrow their activities to their chosen field of specialty (autonomy) [82].

The reasons for the high levels of extrinsic motivation, which we found in junior residents, are primarily because they experience their first professional income (financial factor) [4]. Amotivation occurs when an individual experiences lack of competence and the reason for carrying out an activity and possible outcomes are not identified [9,15]. The need for competence and personal causation are relevant to initiate the identification process with a task and therefore leading to intrinsic motivation [83]. The nature of anaesthesiology residency can be prejudicial for the identification process as it is characterised by low autonomy and low decision-making at the beginning, accompanied by the fact that when emergencies occur, the specialists take over. These circumstances cause the feeling of lacking competence and result in amotivation [14]. Furthermore, at the beginning of residency, the entrusted tasks become mundane and locus of causality and personal causation for these tasks is not perceived from the inside. The required nutriments for the natural process of internalization are lacking and this results in the high levels of extrinsic and relative high levels of amotivation which we found in junior residents.

The prevalence of amotivation in junior residents should not be ignored, as in several studies, junior residents were reported to be at high risk for burnout and emotional exhaustion (stress) [48,49,84].

Our results define an adjusting screw and expand the scope of action to enhance well-being: We found that amotivation was negatively correlated with job satisfaction. Hereby, we complement previous findings, that suggest protective features of job satisfaction against burnout [45,59,60,63]. Therefore, we can conclude that the decrease of amotivation will lead to job satisfaction and protection from declining mental well-being.

It is the responsibility of supervising consultants and medical directors to guide the junior residents and provide them with nutriments to facilitate the internalisation process (e.g., explaining why a task is important), which will result in decreased amotivation.

Further research needs to focus on the identification, development and validation of possible actions and interventions for the working place, targeting the decrease of amotivation in junior residents.

Some limitations of our study merit consideration. Our results derive from a single centre study, assessing the situational motivation of anaesthesiologists from one department. Furthermore, the sample size for each subgroup is small, which could be a potential source of bias and question the generisability of our results. Nevertheless, to our best knowledge, no evidence about the expression of motivation in the field of anaesthesiology or other healthcare settings with a focus on the course of motivation during residency has been published. Neither has any published investigation worked out, if in healthcare, specific types of motivation are necessary based on job characteristics and if some occupational groups are at risk to suffer from all the described disadvantages of lacking autonomous regulation. Therefore, our study can be handled as a piloting work in order to stimulate further investigations. High risk medical workplaces, like anaesthesiology, have several natural job characteristics which influence employees’ motivation. Next to the socioeconomic, organisation and personal factors, these factors need to be identified. Only then motivational changes can be understood entirely, and interventions can be considered. A further limitation is that the study design was based on voluntary participation, which might have only led to the participation of a priori motivated anaesthesiologists, resulting in biased results. However, there is no way to rule out this speculation as these kinds of studies can only be conducted voluntarily.

One might argue, that the use of a pre-existing academic motivational scale [85] might have matched better for our investigation. Our aim was not to measure overall motivation, but rather the situational motivation, because the situational motivation focusses on the actual nature of motivation and equates the operationalization of motivation, regarding the “why” of behaviour [67].

The generalisation of our findings is hard to evaluate, as different departments might provide various environmental influences on daily work tasks of anaesthesiologists. Work motivation is largely influenced by the social context in which employees operate and their motivation is not only affected by resources but also by job demands [34]. Nevertheless, we can derive some implications from our study for clinical practise and as these implications are based on the principles of SDT, they can be transferred to other fields and professions in healthcare.

## 5. Conclusions and Implications

The German translation of the SIMS showed a good applicability and internal validity in the field of anaesthesiology and is therefore suitable to assess employees’ situational motivation, in order to be able to interfere and to foster their well-being and job-satisfaction. The use of the SIMS can be considered for further international studies on healthcare providers’ motivation. The uniform use of one assessment tool would facilitate the comparability of findings.

In summary, our results show that for job satisfaction, intrinsic motivation and autonomous regulation are necessary, but the presence of controlled regulation and extrinsic behavioural regulation do not have declining effects. Controlled regulation is not primarily unsolicited in experts and its expression might rather be required and helpful to master mundane and not outrageously challenging working tasks. Therefore, when controlled regulation coexists with the other motivational qualities, the results will be job satisfaction and better performance.

Our results further indicate that amotivation decreases job satisfaction. As job satisfaction has been suggested to offer protection against burnout, our study points out a new integral relationship between burnout and amotivation. We therefore strongly recommend the decrease of health care workers’ amotivation. The resulting benefits will be extensive and not only affect employees’ well-being, because it has been proven that job satisfaction of health care providers is associated with decreased treatment costs, content patients and better work processes. Therefore, even from an economic point of view, the decrease of amotivation and increase of autonomous regulation is profitable. As amotivation occurs when the reason to carry out an activity is not identified and when an individual experiences a lack of competence or autonomy or relatedness, the decrease of amotivation could be equated to the increase of identification with the task. This identification will result in higher autonomous motivation and this in turn—based on our results—will enhance job satisfaction and well-being.

Therefore, autonomy supportive interventions, with the aim to decrease amotivation should be considered and implemented on various levels:

Healthcare policy should evaluate reforms and budgets of the healthcare system, as health care institutions are often under pressure of cost and time considerations. This pressure creates institutional structures, which by no means support individual development and autonomy of employees. Even if many institutional structures cannot be changed, hospital managements should rethink and adapt their predefined goals and the provided working circumstances.

Hospital managers, in cooperation with clinic or institutional directors, should monitor employees’ motivation in order to interfere and prevent unintended developments. Hereby we recommend the application of the SIMS in order to analyse working circumstances with regard to motivation, instead of large surveys that mainly target to detect the current state.

Medical directors should reorganise institutional policies (as far as possible) creating autonomy supportive working environments and integrate interventions to promote the satisfaction of the three basic psychological needs (autonomy, competence, relatedness) of their employees. One approach can be regular feedback based on employees’ performance and open conversations about why certain things have to be implemented or why some working processes are necessary.

During these considerations, it should not be forgotten that employees also bear the responsibility for their mental health. Therefore, they should expand their coping strategies and accept that certain standards cannot be changed instead of complaining about every shortcoming. A different angle of view, resulting from inner change and acceptance might be more helpful and lead to well-being.

Further studies need to investigate how the mentioned implications, that aim to enhance job satisfaction and decrease burnout, can be translated effectively to healthcare. First, international studies need to explore the motivational course of different specialties in different work settings. Then, interventions based on the mentioned implications (e.g., autonomy supportive feedback) should be systematically developed and their outcomes tested with qualitative and as well with quantitative research methods (e.g., SIMS).

This approach will prevent interventions to be too general and not tailored to the needs of each occupational subgroup.

## Figures and Tables

**Figure 1 healthcare-09-00262-f001:**
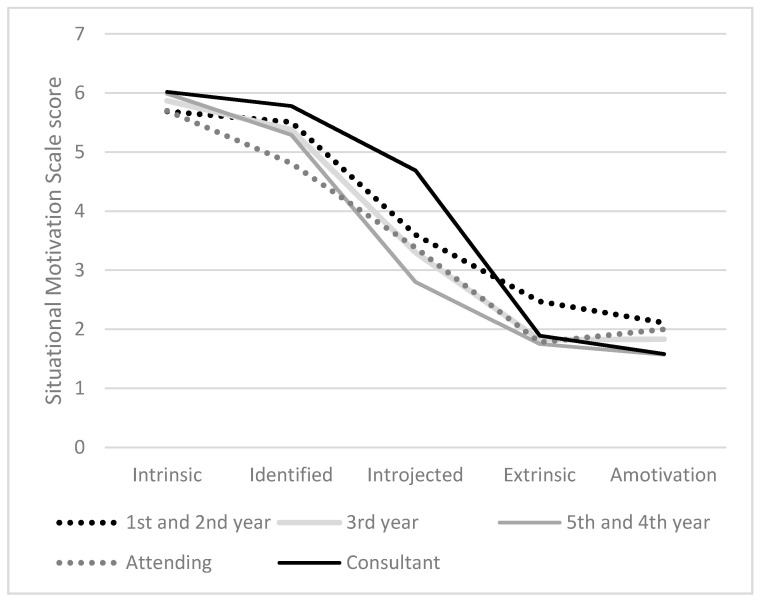
Patterns of motivation reported by the subgroups.

**Table 1 healthcare-09-00262-t001:** Number and anaesthesiology training levels of the study participants.

	1st & 2nd year	3rd year	4th & 5th year	Attending	Consultant
Number of participants	17	15	17	12	10

**Table 2 healthcare-09-00262-t002:** Internal consistency means (standard deviation) of different situational motivational variables and job satisfaction (ANOVA).

Situational Motivation	1st and 2nd Year	3rd Year	4th and 5th Year	Attending	Consultant	ANOVA		
						F (df)	*p*	η^2^
Intrinsic Identified Introjected Extrinsic Amotivation	5.69 (1.23) 5.51 (1.20) 3.60 (1.28) 2.47 (1.56) 2.11 (1.32)	5.87 (0.68) 5.38 (0.88) 3.30 (0.94) ^A^ 1.82 (0.93) 1.83 (0.84)	5.99 (0.77) 5.29 (1.08) 2.80 (1.40) ^A^ 1.75 (0.78) 1.57 (0.72)	5.70 (0.92) 4.81 (1.10) 3.38 (1.04) 1.78 (0.75) 2.00 (0.78)	6.02 (0.44) 5.78 (0.71) 4.69 (0.80) ^A^ 1.89 (0.82) 1.58 (0.51)	0.41(4) 1.31 (4) 4.11 (4) 0.26 (4) 1.01 (4)	0.799 0.277 0.005 ^**^ 0.264 0.407	0.3 0.07 0.20 0.08 0.06
Motivation indices								
Autonomous Controlled	5.60 (1.16) 3.03 (1.10)	5.63 (0.72) 2.55 (0.68)	5.63 (0.79) 2.26 (0.96)	5.26 (0.95) 2.57 (0.63)	5.90 (0.55) 3.29 (0.64) ^B^	0.71 (4) 2.89 (4)	0.586 0.029 *	0.04 0.15
Job satisfaction	5.64 (1.00)	4.91 (0.79)	5.29 (0.77)	4.83 (1.64)	5.78 (0.44)	1.95 (4)	0.112	0.12

Note: * *p* < 0.05; ** *p* < 0.01. The levels of introjected and controlled regulation differed statistically significant in the ANOVA analysis. Then post-hoc analysis with Bonferroni correction for multiple testing was conducted for the significant results. ^A^ Post-hoc analysis revealed that consultants’ levels of introjected regulation was significantly higher than the group of 4th and 5th year residents (*p* = 0.002) and in the group of 3rd year residents (*p* = 0.048). ^B^ Further consultants were significantly higher controlled regulated than 4th and 5th year residents (*p* = 0.045).

**Table 3 healthcare-09-00262-t003:** Zero-order correlations among study variables.

Situational Motivation	2.	3	4.	5.	6.	7.	8.
Intrinsic	0.694 **	0.154	−0.323 **	−0.637 **	−0.79	0.903 **	0.360 **
2. Identified		0.313 **	0.24	−0.354 **	0.23	0.936 **	0.21
3. introjected			0.239 *	0.118	0.826 **	0.262 *	0.28
4. extrinsic				0.401 *	0.745 **	−0.144	−0.218
5. amotivation					0.309 **	−0.523 **	−0.265 *
Motivation indices 6. Controlled regulation							
						0.099	−0.110
7. Autonomous regulation							0.299 *
8. Job satisfaction							

** *p* < 0.01, * *p* < 0.05. The correlations between the subscales of the Situational Motivation Scale and the computed autonomous and controlled motivation confirmed a simplex pattern of relationships across the investigated group of anaesthesiologists. Subscales adjacent along the motivational continuum of self-determination were more positively correlated than more distant ones.

## Data Availability

All data is presented in the manuscript. Further information will be provided by the corresponding author on reasonable request.

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
