# Peer review of "Understanding Why All Types of Motivation Are Necessary in Advanced Anaesthesiology Training Levels and How They Influence Job Satisfaction: Translation of the Self-Determination Theory to Healthcare"

_healthcare, 2021, doi:10.3390/healthcare9030262_

Round 1

Reviewer 1 Report

This article is a very interesting study of the working environment and motivation of anesthesiologists, and a scientific examination of how to motivate them to work in a highly specialized field.

I generally agree with the research design, results, and discussion, but I would like to list some minor points that should be corrected.

  1. Among the many motivation theories, why did we use SDT for a sample of anesthesiologists? The reasons for this should be discussed in detail in the introduction or in the chapter on previous research.
  2. Participants information also appear in Results. This is confusing to the reader, so it should be listed together in the Method. Also, there is no explanation for "38%". It should be stated why the response rate was reduced to 38%.
  3. Situational Motivation Scale: The German translator is listed in the supplementary material; it seems that there are 5 items for each of the 4 subscales, but I could not find that in the German version. The questionnaire content and subscales should be explained.
  4. The author himself has stated in the Discussion as a Limitation, but I was concerned about the small number of samples. A total of 71 people were analyzed by ANOVA by 5 groups, and finally, I think that the significance is derived by Bonferroni adjusted correction. The Note in Table.2 should not only show the results, but also the details including the calculation process.
  5. The author states that "Consultants reported the highest levels of all motivational qualities." not only in Abstract but in some places.
    Perhaps it is shown by the result of the consultant's motivation score in Figure 1, but this consultant's motivation should be explained in Discussion in conjunction with Fig.1.
  6. Due to the paucity of previous studies on anesthesiologists' motivation, I wondered whether it was appropriate to infer the attitudes of anesthesiologists from the questionnaire of the author's department, even though it was a pilot study. The fact that there was a single responding department should also be mentioned in Limitation.
  7. Is there a possibility that there are unique characteristics of anesthesiology compared to other medical specialties and nursing professions in the health care field? In addition, mention should be made of the possibility that the results of the present study may be applicable to departments other than anesthesiology and to other health care professions.

Reviewer 2 Report

I have recommended that the paper is accepted but subject to addressing the comments below: 

While the abstract reads well, reporting more findings and recommendations are needed.

I can see the aim in the introduction but you need to add objectives or research questions that helped you meet the aim. I would like to see how the literature globally, then in Europe, then Germany.

The values of Cronbach Alpha test for the scales need to be reported in the methods. I am missing points in ethical considerations. Did you obtain the relevant approval? If yes, where? what is the number? what steps did you take in line with this in the study?

In the discussion, you reported 'Although one expects Junior residents (1st and 2nd year) to be intrinsically motivated -, in our study, they reported the highest levels of amotivation and extrinsic motivation'. can you expand of the difference between your expectation and the findings. Also, why did you expect this? Provide further discussions on why the there may be the difference and recommend specific studies that can be conducted in future to address this.

The conclusion is too brief. You need to expand on it. Also, I would like to see the implications of the findings for research, practice and society here or in the discussions. What are the implications of this study for the international readership and practice. The recommendations for various parties, academics, professional, organisations and the state needs to be reported.

Round 2

Reviewer 1 Report

I think this is a polite and appropriate response to the previous point. I agree to publish this article as is.